# Comparing the Trail Users with Trail Non-Users on Physical Activity, Sleep, Mood and Well-Being Index

**DOI:** 10.3390/ijerph17176225

**Published:** 2020-08-27

**Authors:** Abbas Smiley, William Ramos, Layne Elliott, Stephen Wolter

**Affiliations:** 1Westchester Medical Center, New York Medical College, New York, NY 10595, USA; Abbas.Smiley@WMChealth.org; 2Recreation Park, and Tourism Studies Department, Indiana University School of Public Health-Bloomington, Bloomington, IN 47404, USA; wramos@indiana.edu; 3Eppley Institute for Parks and Public Lands, Indiana University School of Public Health-Bloomington, Bloomington, IN 47404, USA; laynelli@indiana.edu

**Keywords:** trail, perceived wellness and health, access to nature, outdoor scenery, sleep, physical activity

## Abstract

*Background:* The current study sought to understand whether trail users reported better wellness and health status compared to the non-users, and to recognize the associated factors. *Methods:* Eight trails from different locations and settings within Indiana were selected to sample trail users for the study. Additionally, areas surrounding these eight trails were included in the study as sample locations for trail non-users. Trail users and non-users were intercepted and asked to participate in a survey including demographics, socioeconomic status, physical activity, mood, smoking, nutrition, and quality of sleep. Information was collected and compared between the trail users and the non-users. Association of self-rated health, age, sex, race, marital status, employment, income, education, smoking, nutrition, sleep, and mood with trail use was evaluated by multivariable linear regression model. *Results:* The final sample size included 1299 trail users and 228 non-users. Environmental factors (access to nature and scenery) were important incentives for 97% and 95% of trail users, respectively. Age, sex, mood, and sleep quality were significantly associated with using the trail. Mean (SD) self-rated wellness and health out of 10 was 7.6 (1.4) in trail users and 6.5 (1.9) in non-users (*p* < 0.0001). Importantly, trail users were significantly more physically active outside of the trail compared to the non-users (207 vs. 189 min/week respectively, *p* = 0.01) and had better sleep qualities and mood scores. Using the trails was significantly associated with higher self-rated wellness and health score. The longer the use of trails, the higher the self-rated wellness and health index (β = 0.016, *p* = 0.03). *Conclusion:* Compared to not using the trails, trail use was significantly associated with more physical activity, better sleep quality, and higher self-rated wellness and health.

## 1. Introduction

Well-being stems from socioeconomic status, education, mood status, and several lifestyle factors including sleep, physical activity, nutrition, social support, connection to nature, and individual habits such as smoking [1,2,3,4,5,6,7,8,9,10,11,12,13,14,15,16,17,18]. On the other hand, inadequate knowledge, improper attitude, and inappropriate practice such as unbalanced sleep, lack of physical activity, poor nutrition and lack of recreation are the mainstems of obesity epidemics [19,20,21,22,23,24,25,26,27,28,29]. As more than one-third of Indiana population is obese, Indiana State implemented the Hoosiers on the Move program with the objective to build new trails that provide pedestrian-biking trail access within 9 km for every Indiana resident by 2020, [30] with the greater goals to help economic growth, promote health and wellness, and to reduce the increasing rate of obesity in Indiana. Indiana currently offers more than 6500 km of trails, which is more than double the mileage existing 10 years ago [31]. Indiana’s goals and objectives were based on studies that have shown that access to nature and outdoors [3,4,32,33] and building new trails increase the amount of physical activity in terms of walking, running, and biking. They also grow the future intention to be more physically active in people living close to the trail compared with people living far from the trail [34,35,36,37]. Similarly, research shows starting to use trails is a good habit that increases the amount of physical activity for individuals [37,38]. In addition, several studies have shown the direct relationship between physical activity and well-being among various age groups of the population [18,39,40,41,42]. However, the positive effects of using the trails are beyond the benefits of physical activity in trails. For example, connection to nature and social interactions are among some major benefits [3,4,32,33]. The literature is unclear if using trails boosts the wellness and health status for trail users. The current study tried to document this relationship. The goal of the current study was to assess whether trail users reported better wellness and health status than the trail non-users.

## 2. Methods

The 2017 Indiana Trails Study was carried out on Indiana State trails. Details of the methods were explained elsewhere [43,44]. In summary, data on demographics, socioeconomic status (SES), physical activity, mood, smoking, sleep, and diet were collected and compared between the users of trails and the non-users. Physical activity and trail use were assessed through Recreation Trail Evaluation Survey (RTES) [45]. RTES is a valid and reliable tool, with 34 time scale, multiple choice, or Likert scale questions, that inquires about the trail use in terms of social and chronological patterns, type, time, and the amount of physical activities performed both inside and outside of the trail, and the attitudes, safety, accessibility, and concerns about the trails. For instance, it asks: What type of activity do you usually do on the trail? Walking, running, biking, and other types of physical activity are multiple choices for this question that participants were able to select from. Diet [46,47] and mood [11] were evaluated according to the Gallup Diet Questionnaire and Gallup Well-Being Index. They were asked to report number of days in a week that they had fast food, less than 4/5 serving of fruits/vegetables, sadness, no energy to get things done, anger, worry, and physical pain. Each diet or mood question had a score of 0 (never) to 7 (every day of the week). The sum of scores of diet questions represented the total diet score and the sum of scores of mood questions represented the total mood score. The higher the scores, the worse the diet and the mood. Trail user opinions and use factors were determined using survey questions consistent with the past trails’ studies [48]. Sleep was evaluated by Mini-Sleep Questionnaire [49]. Each question scored 0 (never) to 7 (every day of the week) and the sum of all questions represented the sleep score in regression analysis. The higher the sleep score, the worse the sleep quality. Self-rated wellness and health (10-point scale) were determined as the last question of the survey, with 10 being the healthiest state and 1 the unhealthiest [50,51,52].

Ethics approval and consent to participate: The Office of Research Compliance at Indiana University approved the study protocol data collection. All subjects consented to participate in the study (No. 1606065577).

### 2.1. Trail User and Non-User Probability Sampling Process

The random selection of subjects and trails across Indiana was a critical first step in the research. Using a multi-stage sampling method to address multiple locations in trails, differing locations on and near the selected trails, and times and days of the week that users may prefer were carefully managed to avoid the problems of self-selection sample bias. Additionally, data collection using organization volunteers who would not be appropriately trained in the protection of human subject required detailed sampling processes based on multiple factors in a multi-stage probability sampling process. Factors considered and managed during sampling included weather patterns, regions (Northern, Central, and Southern Indiana), land use (urban, suburban, and rural), agency capacity factors, trail counters’ access, and access to the list of neighboring properties.

The volunteers were located at predefined trailheads and distributed the research information and the relevant online link to the survey. Data gathering was executed in the second week of April, June, and August and the first week of October, from 6 a.m. until 8 p.m. or dusk if it was before 8 p.m. The selection of participants in the study was carefully planned to best meet the standards of a probability sample as the study’s budget did not allow for placing appropriately human subject-trained researchers at each trail and non-trail site selected. The resulting use of volunteers to collect data across the state complicated the sample selection process, resulting in rigorous processes to vary the intercept location, day of the week, and time of the day for both trail and non-trail sites. Trained volunteers were assigned to intercept and ask trail or non-trail potential study subjects, using a greeting–introduction–question answering protocol, to participate in the study [45]. Trained volunteers distributed postcards directing selected participants to a website URL containing survey information or a phone number to call to receive a mail copy with self-addressed, stamped return envelope to interested subjects. Potential trail non-user subjects were offered an incentive ($5 Amazon gift card) to participate.

### 2.2. Trail Selection

The trail selection criteria utilized were distribution of trails among urban, suburban, and rural setting (based on the predominant surrounding land use), trails with an equal mix of each desired, distribution of study trails in the Northern, Central, and Southern geographic regions of Indiana (as defined by the Indiana Department of Transportation), and trails with a viable organization operating the trail agreeing to be a partner in collecting data at the trail using volunteers (Figure 1) [53].

### 2.3. Trail User and Trail Non-User Survey Implementation

As discussed earlier, the survey was implemented by stationing trained volunteers from the trail management organization at specified locations, specified weeks (one week each month from April to October 2018), and times and days during the study period so they could give out the relevant information about the study, such as online trail survey link. An incentive of a $5 gift card was considered for trail non-users to complete the survey. The volunteers gave the relevant information and cards to invite every participant at the survey locations.

To intercept trail users at the start or end of trail use, popular trailheads were chosen. To select the control group, locations were selected from the same area of the trail that were frequently used by the community, such as grocery stores or libraries.

### 2.4. Statistical Methods

Chi-square test was used to evaluate the index distributions of demographics, socioeconomic status, sleep, nutrition, and mood patterns between the two groups. Univariable association of all variables and trail use was evaluated through logistic regression analysis. The probability of using trail vs. non-using the trail was evaluated by multivariable logistic regression, controlled for sex, age, race, employment, income, marital status, education, smoking, nutrition, sleep, and mood. Adjusted odds ratios (ORs) and the confidence intervals (CIs) were reported.

Self-rated wellness and health were continuous variables presented as mean and standard deviation (SD). Univariable association of self-rated wellness and health with every variable was also evaluated through linear regression analysis. Multivariable linear regression adjusted for confounders, such as age, sex, race, marital status, employment, income, education, smoking, nutrition, sleep, and mood, were performed to determine the association between self-rated wellness and health and trail use. There were three reasons behind the selection of confounders, according to literature review. First, they are associated with self-rated wellness and health. Second, they are associated with the trail use. Third, they are located in the causal path from the trail use to wellness and health. In order to identify the useful subset of the predictors and reduce the multicollinearity problem and to resolve the overfitting problem, backward elimination process was used. Data analyses were conducted using SPSS program (SPSS, Version 26, Chicago, IL, USA) and *p* < 0.05 was considered significant.

## 3. Results

The final sample size included 1299 trail users and 228 non-users. Demographic characteristics and SES of trail users vs. trail non-users are demonstrated in Table 1. They were significantly different in terms of sex, age, race, SES, and marital status distribution. About 56% of trail users vs. 38% of non-users were males. People older than 45 years composed 65% of users and 45% of non-users. This explains the higher percentage of retired and married people among users vs. non-users, plus the lower percentage of students and smoking among users vs. non-users (Table 1).

Distribution of extreme sleep, mood, and diet patterns are compared between the two groups in Table 2. There were significant differences in distribution of all items between the two groups. Fifty percent or more of trail users never experienced 7/9 negative sleep symptoms including difficulty falling asleep, waking too early, falling asleep during the day, snoring, headache on wake up, excessive daily sleepiness, and excessive sleep movement (Table 2), whereas 50% or more of trail non-users never experienced only 3/9 negative sleep symptoms (falling asleep during the day, headache upon waking, and excessive movement during sleep). Similarly, more than 50% of the trail users never experienced 4/5 negative mood symptoms of lack of energy, sadness, anger, and physical pain, whereas the corresponding prevalence of never experiencing these symptoms was below 50% for the trail non-users (Table 2). Again, more than half of the users never ate fast food, whereas only one-third of the non-users followed this dietary habit.

The main physical activities in trail users were walking, running, and biking. These three categories were similar in most characteristics such as age, race, education, and income. Of walkers, runners, and bikers, 62, 63, and 60% were in the 36–65-year-old group, respectively. More than 90% in all three categories were White. At least 75% had college graduate within each category, and 10–14% had household income less than $38,000 per year. Their sex distribution was somehow different; 60, 41, and 36.5% were female, respectively. Walking, running, and biking were reported by 29, 19, and 52% in trail users, respectively, vs. 73, 22, and 21% in trail non-users, respectively. This means trails non-users were more active in terms of walking, similar to the trail users in terms of biking, and less active in terms of biking. Other than walking, running, and biking, strength training and gardening were the most frequent types of physical activity reported by 39 and 37% in trail users, respectively, vs. 35 and 32% in trail non-users, respectively. The frequency distribution of all other types of physical activities, such as swimming, aerobic dance, yoga, martial arts, racquet sport, golfing, and team sport, were similar between the two groups and were reported by 1–15% in both groups.

Trail users were asked to indicate whether the amount of their physical activity level has increased, decreased, or stayed the same since they started to use the trail. More than two-thirds answered Increased (Table 3). Then, those who answered that their physical activity has been increased or decreased were asked to indicate how much their physical activity has changed since they started to use the trail. Table 4 shows that about three-fourths of those who experienced increased physical activity had more than 25% higher amount of physical activity since using the trail. Moreover, trail users were compared with the trail non-users in terms of the amount of time spent on physical activity per week. Trail users were significantly more physically active outside of the trail vs. the trail non-users, 207 vs. 189 min/week, respectively (*p* = 0.01). Physical activity of less than 2.5 h/week, 2.5–5 h/week, and more than 5 h/week were reported by 33, 34, and 33% of trail users, respectively, vs. 39, 34.5, and 26.5% of non-users, respectively. These findings altogether mean using the trails was associated with being more active.

Factors that increased trail use were asked from the trail users through RTES questionnaire. Among them, outdoor activity factors were the most prominent; 95% and 97% of trail users reported access to scenery (beauty of environment) and access to nature/environment as the important factors to their use of trails, respectively (Table 5).

Table 6 shows the univariable association of every variable and the trail use in terms of OR and 95% CI. Almost all variables had significant univariable association with using the trail. However, when all were employed in multivariable logistic regression, only age, sex, sleep, and mood remained significant.

Mean (SD) self-rated wellness and health out of 10 was 7.6 (1.4) in trail users and 6.5 (1.9) in non-users (*p* < 0.0001). Table 7 demonstrates the univariable association of every variable with the self-rated wellness and health. Nine variables had significant univariable association with self-rated wellness and health. When all variables were entered in multivariable linear regression, six of them remained significant, which included age, smoking, nutrition, sleep, mood, and using the trails. In other words, using the trail predicted a higher wellness and health score, whereas lower sleep quality and inferior nutrition were associated with lower wellness and health score (Table 7).

Finally, multivariable linear regression model was employed among only the trail users to evaluate the association of self-rated wellness and health index and the years of trail use. Interestingly, the model showed a significant positive association (β = 0.016, *p* = 0.03) between them after controlling for other variables mentioned in Table 7; the longer the use of trails, the higher the self-rated wellness and health index.

## 4. Discussion

### 4.1. Sociodemographic Factors

The current study compared the wellness and health status between the trail users and trail non-users. Among the walkers, educated, married, employed, middle-aged women revealed the highest frequency of walking. Among the runners, educated, married, employed, young/middle-aged men showed the highest frequency of running. Among the bikers, educated, married, employed, middle-aged men demonstrated the highest frequency of biking. Parallel outcomes have been observed by similar studies. For example, educated, employed, middle-aged women showed the highest frequency of walking in Missouri [38]. Educated, married, employed, middle-aged men showed the highest frequency of biking in Australia [54]. Age, sex, mood, and sleep quality were associated with using the trails in the current study. The higher the age, the higher the probability of using the trails.

### 4.2. Behaviors

Bad sleep and negative mood decreased the probability of using the trails. Given that the trail users were significantly older than the non-users, they were expected to have more frequent negative sleep symptoms, whereas the opposite was observed in our study; i.e., trail users reported markedly better sleep qualities. The interactive relationship among health/wellness, physical activities, and sleep have been shown or explained by several studies [9,10,14,15,55,56].

### 4.3. Wellness and Health

The study showed that perceived wellness and health among the trail users were significantly higher than that of the trail non-users. The benefits of using the trails are not limited to increased physical activity, better sleep qualities, and more stable moods. Access to nature, scenery, and beauty of environment were important incentives for almost all trail users in our study. The significant human restorative effects of connection to nature has been demonstrated in several investigations [3,4,32,33,57]. The recreational benefits and the biodiversity of the environment have significant psychological advantages [58,59,60,61,62,63]. Biodiversity of environment refers to living organism variability of the environment. Regular use of the trail combines all the above-mentioned advantages: Physical activity, sleep, nature, and social activity. Then, increased physical activity, restoration, psychological well-being, and recreation are the consequences. Low physical activity has an established causal role in obesity and anxiety/depression. Then, our findings provided some justification for the possibility of constructing the health and wellness through trail activities and highlighted the importance of building additional trails throughout the country.

### 4.4. Limits and Strengths

This study contained some limitations. Its cross-sectional design did not lead to establish a causal role of type of physical activity in self-rated wellness and health. The sample size was different between the two groups. This was overcome by employing proper statistical analyses and using appropriate number of predictors to determine the differences between trail users and non-users. We also proposed a small financial incentive. However, knowledge of the financial incentive did not significantly increase the response rate. That is why we believe it did not bias the response of participants. The nutritional patterns of participants were controlled roughly by two questions about fruits/vegetables and fast food. Clearly, having other relevant information of diet could have improved the related model adjustment. Additionally, specific times were selected to recruit participants, whereas the actual volunteer participation could have been different. A strength of our study was the comparison of self-rated wellness and health of trail users with that of trail non-users, controlling for important confounders such as demographics, SES, mood, and lifestyle. Furthermore, efforts to diminish the recall bias in seasonal variations was undertaken by assessment of the trail users in four seasons. Follow-up investigations may improve the generalizability and the reliability of our outcomes.

## 5. Conclusions

Our study showed that perceived wellness and health among the trail users were significantly higher than that of the trail non-users, and that physical activity, sleep, and nature were the prominent features in promoting the use of the trail by participants. Specific health outcomes related to sleep, restoration, psychological well-being, and overall health are perceived to be greater in trail users.

## Figures and Tables

**Figure 1 ijerph-17-06225-f001:**
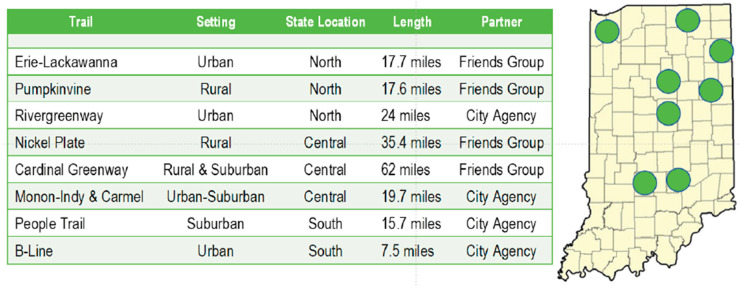
The map of selected trails for 2017 Indiana Trail Study.

**Table 1 ijerph-17-06225-t001:** Demographics and socioeconomic status information of trail users vs. trail non-users.

Demographic & SES Characteristics		Trail Users		Non-Users		*p* Value
		Frequency	Percentage	Frequency	Percentage	
Age, years	18–25	78	6	50	22	0.0001 *
	26–35	169	13	39	17	
	36–45	208	16	35	15.5	
	46–65	584	45	68	29.5	
	>65	260	20	36	16	
Sex	Female	571	44	139	61	0.0001 *
	Male	728	56	89	38	
Race	White	1214	93.5	205	90	0.015*
	Black	20	1.5	11	5	
	Hispanic	39	3	7	3	
	Asian	20	1.5	3	1.5	
	Indian	6	0.5	1	0.5	
Marital Status	Single	240	19	69	30.5	0.0001 *
	Married	929	71.5	128	56	
	Widowed	26	2	8	3.5	
	Divorced	104	8	23	10	
Employment	Homemaker	45	3.5	14	6	0.0001 *
	Self-employed	131	10	23	10	
	Student	52	4	30	13	
	Employed	740	57	121	53	
	Retired	318	24.5	39	17	
	Not Employed	13	1	2	1	
Job Satisfaction	<30%	65	5	32	14	0.001 *
	30–70%	273	21	34	15	
	>70%	961	74	162	71	
Education	<9th Grade	13	1	1	0.5	0.0001 *
	High School	189	14.5	69	30	
	Technical School	84	6.5	6	2.5	
	College Graduate	520	40	86	38	
	Graduate School	344	26.5	45	20	
	Professional Degree	149	11.5	21	9	
Household Income	<$10,000	26	2	16	7	0.0001 *
	$10,000–38,000	130	10	48	21	
	$38,001–91,000	546	42	86	38	
	$91,001–190,000	454	35	64	28	
	>$190,000	143	11	14	6	
Smoking	Yes	39	3	17	7.5	0.004 *
	No	1260	97	211	92.5	

* *p* value less than 0.05.

**Table 2 ijerph-17-06225-t002:** Sleep, mood, and eating patterns compared between trail users and non-users.

How Many Days Per Week, Do You Have:	Never, %		Everyday, %		*p* Value
	Trail Users	Non-Users	Trail Users	Non-Users	
Difficulty Falling Asleep	54	41	2	9	0.0001 *
Too Early Wake Up	50	43	4	9	0.001 *
Hypnotic Medications Use	89	82	3	6	0.001 *
Falling Asleep During Day	68	55.5	1.5	2.5	0.007 *
Tired Feeling Upon Waking	19	38	4.5	16.5	0.0001 *
Snoring	62	49	12.5	18	0.01 *
Mid-sleep Awakenings	34	22.5	18.5	24.5	0.02 *
Headache Upon Waking	83	68	0.1	2	0.0001 *
Excessive Daytime Sleepiness	58.5	34.5	2	6	0.0001 *
Excessive Movement During Sleep	72	59	3.5	9.5	0.0001 *
Lack of Energy	61	36.5	1	5	0.0001 *
Sadness	68	49	2.5	3	0.0001 *
Anger	61	44	1.5	7.5	0.0001 *
Physical Pain	55	34.5	8.5	17	0.0001 *
Worry	48	31.5	6	13	0.0001 *
Fast Food Meals	51	33.5	0.5	0.5	0.0001 *
<5 Servings of Fruits & Vegetables	29	16.5	9	14	0.0001 *

* *p* value less than 0.05.

**Table 3 ijerph-17-06225-t003:** The change of physical activity since the beginning of trail use, reported by trail users.

Value	Frequency	Percentage
Increased	860	66.6
Decreased	14	1.1
Do not know	24	1.9
Stayed the same	393	30.4
Missing	8	
Total	1299	

**Table 4 ijerph-17-06225-t004:** The amount of increased physical activity reported by trail users (since starting trail use, reported by trail users).

Increased Activity Reported	Frequency	Percentage
0–25%	226	27.3
26–50%	306	37.0
51–75%	133	16.1
76–100%	99	12.0
Over 100%	63	7.6

**Table 5 ijerph-17-06225-t005:** Reasons for trail use instead of other facilities.

Importance	Scenery (Beauty of Environment)		Outdoors (Access to Nature/Environment)	
	Frequency	Percentage	Frequency	Percentage
Least Important	10	0.8	12	1
Somewhat Important	61	4.9	29	2.4
Important	227	18.3	107	8.7
Quite Important	423	34.1	305	24.8
Most Important	518	41.8	778	63.2
Missing	60		68	
Total	1299		1299	

**Table 6 ijerph-17-06225-t006:** Backward logistic regression analysis showing univariable and significant multivariable associations of predictors of not using the trails. The dependent variable was using vs. non-using the trails. All other variables were considered as independent variables.

Predictors		Univariable Association		Multivariable Full Model, R^2^ = 0.183		Multivariable Final Model, R^2^ = 0.159	
		OR (95% CI) **	*p*	OR (95% CI) **	*p*	OR (95% CI) **	*p*
**Age, Years**		1.49 (1.33–1.67)	0.0001 *	1.43 (1.18–1.74)	0.0001 *	1.39 (1.16–1.65)	0.0001 *
Sex, Male		2.00 (1.49–2.69)	0.0001 *	2.03 (1.43–2.89)	0.0001 *	1.98 (1.41–2.77)	0.0001 *
Race	White	0.99 (0.41–2.42)	0.997	0.68 (0.26–1.78)	0.450	Race, Marital Status, Employment, Income, Education, Smoking and Nutrition Were Removed By Backward Elimination	
	Black	0.26 (0.08–0.82)	0.021 *	0.18 (0.05–0.66)	0.010 *		
	Alaska	0.35 (0.03–4.53)	0.424	0.20 (0.01–2.90)	0.240		
	Asians	0.76 (0.17–3.52)	0.730	0.49 (0.09–2.72)	0.420		
Marital Status	Married	0.76 (0.44–1.30)	0.316	1.18 (0.63–2.21)	0.125		
	Widow	1.59 (0.97–2.63)	0.068	1.74 (0.85–3.54)	0.610		
	Divorce	0.66 (0.26–1.68)	0.0391 *	0.74 (0.24–2.26)	0.600		
Employment	Self-Employed	1.87 (0.68–5.14)	0.222	1.36 (0.40–4.6)	0.620		
	Student	3.51 (1.40–8.78)	0.007 *	1.83 (0.61–5.46)	0.280		
	Employed for Wages	1.06 (0.42–2.67)	0.894	1.59 (0.49–5.09)	0.435		
	Retired	3.77 (1.67–8.52)	0.001 *	2.54 (0.94–6.89)	0.065		
	Not Employed	5.27 (2.22–12.50)	0.0001 *	1.63 (0.56–4.75)	0.370		
Income		1.56 (1.33–1.84)	0.0001 *	0.53 (0.70–0.23)	0.10		
Education		1.26 (1.12–1.42)	0.0001 *	1.29 (0.35–4.70)	0.25		
Smoking		0.41 (0.23–0.74)	0.003 *	0.68 (0.32–1.45)	0.320		
Nutrition		0.89 (0.86–0.94)	0.0001 *	0.97 (0.91–1.03)	0.30		
Sleep		0.94 (0.93–0.96)	0.0001 *	0.97 (0.95–0.99)	0.004 *	0.97 (0.95–0.99)	0.002 *
Mood		0.92 (0.90–0.93)	0.0001 *	0.96 (0.93–0.99)	0.01*	0.95 (0.93–0.98)	0.001 *

* *p* value less than 0.05; ** OR = odds ratio; CI = confidence interval.

**Table 7 ijerph-17-06225-t007:** Backward linear regression analysis showing univariable and significant multivariable associations of predictors of self-rated wellness and health. The dependent variable was self-rated wellness and health index. All other variables were considered as independent variables.

Predictors	Univariate Association		Multivariable Model, R^2^ = 0.251	
	β (95% CI) **	*p*	β (95% CI) **	*p*
Age, Years	0.32 (0.25–0.39)	0.0001 *	0.24 (0.17–0.31)	0.0001 *
Sex, Male	−0.17 (-0.33–−0.01)	0.04 *	Sex, Race, Marital Status, Employment, Income and Education Were Removed By Backward Elimination	
Race	−0.05 (-1.41–−0.39)	0.30		
Marital Status	0.13 (0.02–0.23)	0.0 2*		
Employment	0.02 (−0.06–0.10)	0.65		
Income	0.25 (0.16–0.34)	0.0001 *		
Education	0.05 (−0.009–0.11)	0.10		
Smoking	1.16 (0.74–1.57)	0.0001 *	0.64 (0.24–1.03)	0.001 *
Nutrition	−0.11 (−0.13–−0.08)	0.0001 *	−0.04 (−0.07–−0.01)	0.003 *
Sleep	−0.05 (−0.06–−0.04)	0.0001 *	−0.01 (−0.02–−0.002)	0.017 *
Mood	−0.10 (−0.11–−0.09)	0.0001 *	−0.07 (−0.09–−0.06)	0.0001 *
Using the Trails	1.07 (0.86–1.29)	0.0001 *	0.59 (0.37–0.80)	0.0001 *

* *p* value less than 0.05; ** CI = confidence interval.

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
