# Peer review of "Comparing the Trail Users with Trail Non-Users on Physical Activity, Sleep, Mood and Well-Being Index"

_ijerph, 2020, doi:10.3390/ijerph17176225_

Round 1

Reviewer 1 Report

A comparative analysis of a number of variables of health and subjective well-being in users and non-users of trails can be useful for assessing the impact of physical activity on the overall well-being of a person. This is an original contribution to the study of health factors. However, the article can be improved in a number of ways.

In the introduction, the authors state the relevance of the topic, but do not provide a brief overview of the problem under study. It is also not clear what type of trail users the article refers to. Probably about the same or different (using the fixtures). Does a positive view of trail users ' practice correlate with previous studies of the level (dose) of physical activity and well-being?

Lines 53-56 it is not clear to me what the content of this questionnaire is, what its metric indicators are. Lines 57-62 are also in need of a more detailed view. You need to specify a particular scale.

Line 88: I think we should clarify: what is the incentive?

Lines 110-122 it would be desirable to place a justification for using this statistical criterion.

It is not clear from tables 6-7 how much of the variation in the dependent variable (self-rated wellness and health) is explained by these predictors. It would also be desirable to describe the method of linear regression analysis.

It was only in the discussion that it became clear that the sample was made up of different types of Trail-Users. Maybe the authors will consider it possible to explain a little: whether there are differences among walkers, runners, bikers by the studied variables.

Another question: for the author, well-being, psychological well-being is the same as mood? It would be necessary to explain what the authors study specifically from these phenomena.

Reviewer 2 Report

General comments

It is an interesting and far-reaching work. However, it lacks more content and better presents the results. Results are presented that are not associated with the objective of the study.

Quotations should be placed between [ ] as indicated by the rules, not in superscript.

The discussion needs more development and order.

Limitations should be added, and the financial incentive given to the “no users” should be highlighted, and how this could affect the responses. Also highlight how the sociodemographic differences of both groups could affect the comparisons.

Introduction

Add information about other trails or bike lines that help you understand their real benefits. With the little information presented, it is not possible to understand the research question.

It is not clear if the use is for walking, cycling or both. More information should be included in this regard.

Methods.

You must add an accurate description of how the sample was obtained.

Specific comments

Line 103. The incentive of five-dollars gift card, could be written in line 88.

Line 110. The first sentence could be deleted, is a little obvious.

Line 124. The results title was badly writing.

Line 144. It is difficult to perform accurate statistics with such a low “n” in the “no users”. There are comparisons with 1 or 2 cases. Review.

Line 145-150. It is necessary to use only one key term, exercise or physical activity, but or mix them when referring to the results, since they are different things.

Line 166. The presentation of the table is not good.

Put the OR values on the same line as CI. I recommend that the table be horizontal.

Also, it is not understood which variables were removed by backward elimination in multivariable final model.

It would be appropriate to carry out predictive analyzes with the group “no users” to compare (in tables 6 and 7), since it is placed in the title and in the objective. That would give more consistency to the article.

Line 184. The discussion in general is poor. I recommend including subtitles that order the ideas. For example, Sociodemographic factors, Behaviors, Wellness and health, etc.

Reviewer 3 Report

Dear Authors,

I think the article is fine, however some suggestions can improve the understanding for readers:

  • To indicate the meaning of all abbreviations: SES, SD and in the right place, for example SES is explained in the results section however has been used before in the method section. It must be explained also in the tables. On the other hand, the meaning of ORs and Cis can be deleted because they are not used in the main document, they are only used in the table, where the meaning of all abbreviations must be indicated.
  • To use units in the international system, kilometers instead of miles: introduction, figure
  • To include a significant symbol (*, #,..) in all tables with the corresponding p value (<0.05, <0.01, <0.001)

Abstract.

  • p=0101 it is not significant
  • The conclusion does not match the title. Give more details in the conclusion or reduce the title to wellness and health status
  • Do not repeat words already in the title

Methods.

  • To add space: “score,the worse”

Results.

  • Regression instead of regeression (table 6)

Reviewer 4 Report

Congratulate the authors for their work, which I'm sure has involved a great deal of effort.

The article discusses a comparison of Physical Activity, Sleep, Mood and Well-being Index, between Trail subjects and non-trail subjects. The article also shows that Trail subjects have been little studied in relation to their study variables. They make correct comparisons, this being one of the strong points of the article. The main elements to be improved and that in my opinion limits it is the decompensation between the number of subjects in the group of non-users with the number of users and that the group of non-users does not have similar characteristics in which to make a categorization. The authors solve this with good analysis of the characteristics of both groups. But I believe that it would be necessary to increase the information especially relative to the physical activity, not only in quantity but if it is possible in type of practiced activities of the group of not users.

The introduction is too specific. Information on the relationship between physical activity and well-being should be expanded and the benefits of Trail on other types of physical activity should be known.

In the discussion the term walkers, frequency of walkings is used repeatedly, but this concept is not clearly related in the results. This aspect should be further clarified to help the reader understand it. Similarly, no results are seen in relation to bikers.

Since the main objective is perceived wellness and health among the trail users, I consider that too much data is given in the results in which those related to the main objective stand out little in table 7 on page 8 of lines 179-183.

In the results it would be interesting to pay more attention to the issues related to "time and the amount of physical activities performed both inside and outside of the Trail" from the Recreation Trail Evaluation Survey (RTES). 24

One of the weaknesses of the investigation is the difference in size between 1299 users and non-users 228 , although this is overcome by good statistical analyses using a good number of predictors to determine the relationship between Trail users and non-users. But I believe that the participation or not of non-users in other physical activities should be used as a predictor. Because it is difficult to draw conclusions about the relationship between Trail participation and participation in other physical activities, or it is the practice of physical activity itself that can establish such differences between the groups.

 The results in line 150-154 on page 5 should be expanded.

Round 2

Reviewer 2 Report

Dear authors.

Thank you for the corrected version.

Lucky with another's reviewers.

Reviewer 4 Report

Congratulations to the authors as they have responded perfectly to the doubts raised in the first review.